# An Early, Universal Mediterranean Diet-Based Intervention in Pregnancy Reduces Cardiovascular Risk Factors in the “Fourth Trimester”

**DOI:** 10.3390/jcm8091499

**Published:** 2019-09-19

**Authors:** Carla Assaf-Balut, Nuria Garcia de la Torre, Alejandra Durán, Elena Bordiu, Laura del Valle, Cristina Familiar, Johanna Valerio, Inés Jimenez, Miguel Angel Herraiz, Nuria Izquierdo, Isabelle Runkle, María Paz de Miguel, Carmen Montañez, Ana Barabash, Martín Cuesta, Miguel Angel Rubio, Alfonso Luis Calle-Pascual

**Affiliations:** 1Endocrinology and Nutrition Department, Hospital Clínico Universitario San Carlos and Instituto de Investigación Sanitaria del Hospital Clínico San Carlos (IdISSC), 28040 Madrid, Spain; carlaassafbalut90@hotmail.co.uk (C.A.-B.); nurialobo@hotmail.com (N.G.d.l.T.); aduranrh@hotmail.com (A.D.); elena.bordiu@salud.madrid.org (E.B.); lauradel_valle@hotmail.com (L.d.V.); cristinafamiliarcasado@gmail.com (C.F.); valeriojohanna@gmail.com (J.V.); i.jimenez.varas@gmail.com (I.J.); irunkledelavega@gmail.com (I.R.); pazdemiguel@telefonica.net (M.P.d.M.); mcmnita@hotmail.com (C.M.); ana.barabash@gmail.com (A.B.); cuestamartintutor@gmail.com (M.C.); marubioh@gmail.com (M.A.R.); 2Facultad de Medicina, Medicina II Department, Universidad Complutense de Madrid, 28040 Madrid, Spain; maherraizm@gmail.com (M.A.H.); nuriaizquierdo4@gmail.com (N.I.); 3Centro de Investigación Biomédica en Red de Diabetes y Enfermedades Metabólicas Asociadas (CIBERDEM), 28029 Madrid, Spain; 4Gynecology and Obstetrics Department, Hospital Clínico Universitario San Carlos and Instituto de Investigación Sanitaria del Hospital Clínico San Carlos (IdISSC), 28040 Madrid, Spain

**Keywords:** Mediterranean diet, pregnancy, metabolic syndrome, postpartum

## Abstract

An early antenatal dietary intervention could play an important role in the prevention of metabolic diseases postpartum. The aim of this study is to evaluate whether an early, specific dietary intervention reduces women’s cardiovascular risk in the “fourth trimester”. This prospective cohort study compares 1675 women from the standard-care group (ScG/n = 676), who received standard-care dietary guidelines, with the intervention group (IG/n = 999), who received Mediterranean diet (MedDiet)-based dietary guidelines, supplemented with extra-virgin olive oil and nuts. Cardiovascular risk was determined by the presence of metabolic syndrome (MetS) and insulin resistance syndrome (IrS) (HOMA-IR 3.5) at 12–14 weeks postpartum. MetS was less frequent in the IG (11.3 vs. 19.3%, *p* < 0.05). The intervention was associated with a reduction in the relative risk of having MetS: 0.74 (95% CI, 0.60–0.90), but not in the risk of IrS. When analyzing the presence of having one or more components of the MetS, the IG had significantly higher rates of having 0 components and lower rates of having ≥1 (*p*-trend = 0.029). An early MedDiet-based nutritional intervention in pregnancy is associated with reductions in postpartum rates of MetS.

## 1. Introduction

Women with gestational diabetes mellitus (GDM) are at an increased risk of developing cardiovascular disease and diabetes in the postpartum period [1]. In these women, immediate postpartum health evaluation is key in the early detection of cardiometabolic disease risk factors [2]. Identifying the presence of metabolic syndrome (MetS) or insulin resistance (IR), with or without impaired glucose regulation, is the earliest opportunity to detect women at cardiovascular risk. This allows the early start of prevention strategies and treatment.

Women who have a healthy pregnancy are rarely evaluated in the postpartum period. Recently, the American College of Obstetricians and Gynecologist has released a statement about the importance of the “fourth trimester”, reinforcing postpartum follow-up and care of all women to reduce maternal morbidity and mortality [3,4].

A recent study compared rates of MetS and insulin resistance three years postpartum in women with and without GDM, diagnosed by the International Association of the Diabetes and Pregnancy Study Groups (IADPSG) criteria. Women with GDM had significantly higher rates as compared to non-GDM women. However, the presence of MetS and insulin resistance in normoglycemic women was of 6.6 and 9.1%, respectively [5]. Similar results were found in a cohort of women evaluated at three months postpartum [6]. Therefore, it is important to perform a postpartum evaluation, irrespective of having previous GDM.

High adherence to the Mediterranean diet has been associated with a reduced risk of gestational diabetes and other adverse pregnancy outcomes [7,8]. The beneficial effects of this dietary pattern have also been shown in healthy, normoglycemic women [9]. In addition, a Mediterranean diet has been associated with the prevention of type 2 diabetes and metabolic syndrome [10]. Whether the beneficial effect of this diet persists in the postpartum period remains unknown. In a postpartum follow-up of the Finnish Gestational Diabetes Prevention Study (RADIEL), women who received a dietary intervention in pregnancy had a lower incidence of glucose impairment than those who did not [11]. It is unknown whether an early, specific dietary intervention can improve maternal health in the fourth trimester.

The aim of the present work is to evaluate the presence of MetS and insulin resistance—as indicators of cardiovascular risk—at 12–14 weeks postpartum in women who followed a Mediterranean diet-based nutritional intervention in early gestation (from 12 gestational weeks) compared to women who followed standard care. In addition, comparisons were made within groups of normoglycemic and women with GDM. 

## 2. Materials and Methods

### 2.1. Study Design

The Hospital Clínico San Carlos provides medical assistance to the core of the Community of Madrid. Pregnant women attend their first gestational visit at 8–12 gestational weeks to get an ultrasound and a prenatal screening test. 

In general, all pregnant women receive nutritional guidelines from their midwives and obstetricians from the start of pregnancy. Before 2016, standard-care guidelines were based on a MedDiet pattern, but limit fat intake, including extra virgin olive oil and nuts.

Universal and centralized screening of GDM (at 24–28 gestational weeks) is performed at the Central Laboratory of the Hospital. The IADPSG criteria have been used to diagnose GDM since 2012. Women with a positive diagnosis are referred to the Diabetes and Pregnancy Unit for treatment and follow-up. Detailed information about the treatment protocol has been published elsewhere [12]. First-line therapy is medical nutrition therapy, based on a MedDiet with a liberalized consumption of extra virgin olive oil (≥40 mL/day) and nuts (a handful a day). Women who do not develop GDM have a usual routine-care follow-up.

From 2016, dietary guidelines given to all pregnant women at the beginning of pregnancy are similar to the ones given as medical nutrition therapy for women with GDM.

This is a retrospective cohort study that clusters women into two groups: the standard-care group and the early-intervention group.

The inclusion criteria were women with normal glucose tolerance (fasting glucose <92 mg/dL) in the first gestational visit (at 8–12 gestational weeks), ≥ 18 years old, single gestation, who attended postpartum evaluation at 12–14 weeks postpartum and who were exclusively breastfeeding at the moment of postpartum evaluation. Exclusion criteria were gestational age at entry >14 gestational weeks, intolerance/allergy to nuts or extra virgin olive oil or any medical conditions or pharmacological therapy that could interfere with the effect of the intervention or compromise the follow-up program.

The studies included in this analysis were approved by the Ethics Committee of Hospital Clínico San Carlos (ethic codes CI 13/296-E and CI 16/442-E) and conducted according to the Helsinki Declaration. All women signed a letter of informed consent.

### 2.2. Study Population

The sample of women used in this cohort study was retrieved mainly from three different studies [7,13,14].

Standard-care group: The women in this group received general advice and leaflets regarding their diet and physical activity, usually provided by the local antenatal clinics. They received standard-care dietary guidelines by midwives based on a Mediterranean diet but limiting the consumption of fats (including extra virgin olive oil and nuts).

From 2013–2015, 1335 women were followed-up, of which 676 (50.6%) attended postpartum evaluation and were exclusively breastfeeding. These women are mainly from the observational study of Ruiz–Gracia et al. [14] and the control group of a randomized controlled trial (St. Carlos GDM Prevention Study) [7].

Early-intervention group: These women received dietary guidelines based on a Mediterranean diet, with emphasis on the consumption of a daily intake of ≥40 mL of extra virgin olive oil and a handful of nuts, provided by dietitians early on in pregnancy (<12 gestational weeks).

From 2015–2017, 1675 women were followed-up, of which 999 (59.6%) attended postpartum evaluation and were exclusively breastfeeding. These women mainly belong to the intervention group of the St. Carlos GDM Prevention Study [7] and an intervention study based on real-world clinical practice [13]. Women from both studies received the same dietary intervention and attended the same gestational visits (at 12–14, 24–28, and 36–38 gestational weeks).

All women who were diagnosed with GDM were followed-up at the Endocrinology Unit and received the same medical nutrition therapy and treatment, irrespective to belonging to the standard-care or intervention group.

### 2.3. Data Collection

All women were evaluated at gestational weeks 24–28 and 36–38, and at 12–14 weeks postpartum. At all visits, blood and urine samples were obtained, and anthropometric measurements and lifestyle were registered.

#### 2.2.1. Clinical Data

The following set of data were collected at baseline: family history of any metabolic disorders (type 2 diabetes mellitus and metabolic syndrome), obstetric history (miscarriages and GDM), educational level, employment, number of prior pregnancies, smoking habits (registering whether they are currently smoking, or they smoked until they discovered they were pregnant) and gestational age at entry (according to the first ultrasound).

#### 2.2.2. Outcomes Measures

The primary endpoint was to evaluate the effect of a Mediterranean diet-based nutritional intervention in early gestation (from 12 gestational weeks) on the presence of MetS (and each of its components) and insulin resistance at 12–14 weeks postpartum. MetS was defined by the National Cholesterol Education Program Adult Treatment Panel III (NCEP-ATP III) criteria. This is having three or more of the following: waist circumference (cm) ≥89.5, fasting plasma glucose (mg/dL) ≥100, and HbA1c (%) ≥5.7, systolic blood pressure (mmHg) ≥130/diastolic blood pressure (mmHg) ≥85, HDL (mg/dL) <50 and triglycerides (g/L) ≥150. IR was defined as having a HOMA ≥3.5, according to cut-off values specified in the Spanish population [15,16]. Fasting glucose (FG) and HbA1c levels were used to determine the presence of prediabetes and diabetes. Prediabetes was classified according to ADA guidelines: A1C 5.7–6.4% and impaired fasting glucose 100–125 mg/dL (5.6–6.9 mmol/L).

Other outcome measures were weight retention, waist circumference, dyslipidemia, blood pressure, and lifestyle. Weight gain was calculated according to self-referred pregestational body weight. Bodyweight change at postpartum (weight retention) was calculated as postpartum weight—pregestational body weight. Height and weight were measured using a digital height measuring scale (Seca 769, seca GmbH & Co. KG, Hamburg, Germany). Waist circumference was measured only in the postpartum evaluation. It was measured with an anthropometric measuring tape, in the horizontal plane between the lowest rib and the iliac crest. This measurement was repeated twice, and the average was registered. Blood pressure, height, weight, gestational weight gain, and body mass index (BMI) were evaluated in pregnancy and at postpartum. The lipid panel consisted of cholesterol (total, low-density lipoprotein (LDL), and high-density lipoprotein (HDL)) and triglycerides. Blood pressure was measured with an electronic, digital sphygmomanometer with adequate armlet after resting 10 minutes in a sitting position (Omron 705IT, Omron Global, Kyoto, Japan).

#### 2.2.3. Lifestyle Assessment

Pregestational, gestational (at 12, 24–28, and 36–38 gestational weeks) and postpartum eating habits were registered. The Diabetes Nutrition and Complications Trial questionnaire were used to evaluate physical activity and general healthy eating habits. It is based on 15 items, three of which evaluate physical activity and the 12 remaining food frequency intakes. This was used to obtain the “Nutrition Score”. The 14-item Mediterranean Diet Adherence Screener was used to evaluate adherence to the Mediterranean diet. It is based on 14 food items and was used to determine the MedDiet score. A detailed description of the evaluation method is described elsewhere [7].

#### 2.2.4. Biochemical Analysis

Blood and urine were obtained after an overnight fast of 8–10 hours. The following data were determined: HbA1c, standardized by the International Federation of Clinical Chemistry and Laboratory Medicine; serum insulin; homeostasis assessment model for insulin resistance, calculated as glucose (mmol/L) × insulin (µUI/mL)/22.7; and fasting glucose. LDL-cholesterol was calculated with the Friedewald formula. Serum levels of HDL-cholesterol were determined by enzymatic immunoinhibition method in an Olympus 5800 (Beckman-Coulter, Brea, CA, USA). Serum level triglycerides were determined using the colorimetric enzymatic method glycerol phosphate oxidase p-aminophenazone (GPO-PAP).

An External Quality Guarantee Program of the SEQC (Sociedad Española de Química Clínica) monthly evaluates the quality of the methods.

### 2.4. Statistical Analysis

Categorical variables are presented with their frequency and percentage distribution, and continuous variables by their mean and standard deviation (± SD). Comparisons between groups for categorical variables were evaluated using the χ^2^ test or Fisher’s exact test. For continuous variables, measures were compared with Student’s t-test. The differences of mean values between postpartum and pregestational for MedDiet score and Nutrition score are given as 95% confidence interval (CI).

The magnitude of the association between study groups and binary outcomes (MetS and each component and IrS) were evaluated using the relative risk (RR) and 95% CI.

All *p* values are two-tailed at less than 0.050. Analyses were done using SPSS, version 21 (SPSS, IBM, Chicago, IL, USA).

## 3. Results

Table 1 shows socio-demographic characteristics of 1675 analyzed women at 12–14 weeks postpartum: standard-care group 676/1335 (50.6%) and intervention group 999/1675 (59.6%). Women in the intervention group had significantly higher rates of family history of MetS, history of miscarriages, lower prepregnancy body-weight and body mass index and had significantly higher employment rates. GDM rates (%) were significantly lower in the intervention group (49.5 vs. 19.9, *p* = 0.001).

Table 2 shows pregestational, gestational and postpartum Nutrition and MedDiet scores of women in the standard-care and intervention group. In both groups, the scores were better postpartum compared with pregestation. When comparing the intervention and standard-care group, postpartum MedDiet scores were better in the intervention group, especially in those who had GDM.

Table 3 compares clinical and biochemical data of all women of the standard-care and intervention group and subdivided by glucose tolerance. Levels of FG, HbA1c, triglycerides, and blood pressure were significantly lower and HDL significantly higher in the intervention group. This was similar in women with GDM, who had significantly lower levels of FG, HbA1c, blood pressure and waist circumference and significantly higher HDL. The intervention group had lower pregravid bodyweight, BMI, weight gain, and postdelivery BMI (all *p* < 0.05).

Figure 1 shows rates of MetS components and anthropometric parameters of all analyzed women and subdivided by subgroups of glucose tolerance. Regarding glucose regulation, less women (%) in the intervention group had fasting glucose ≥ 100 mg/dL and HbA1c ≥ 5.7% (*p* < 0.001) (Figure 1a). Rates of Systolic and diastolic blood pressure (mm Hg) ≥ 130 and ≥ 5 (Figure 1b), HDL-cholesterol < 50 mg/dL and triglycerides ≥150 mg/dL (Figure 1c) were also lower in the intervention group. When analyzing women with GDM, the intervention group had these same results except for triglyceride levels. In addition, women in the intervention group had a lower BMI postpartum, and a higher percentage of them had weight retention <0 kgs, in both GDM and non-GDM women (Figure 1d).

Figure 2a shows the rate of MetS and IR of all analyzed women, comparing standard-care and intervention group. MetS was less frequent in the intervention group (11.3 vs. 19.3, *p* < 0.05). The RR of having MetS was of 0.74 (95% IC, 0.60–0.90) in this group of women. No significant differences were observed for neither IrS (0.88 (0.70–1.11)) nor for each component of MetS analyzed individually (data not shown). When comparing the rates of having from 0 to 6 components of MetS, the intervention group had significantly higher rates of having 0 components and lower rates of women having ≥1 components (*p*-trend = 0.029) (Figure 2b).

Figure 2c compares the rates of MetS and IrS, and components of MetS subdivided into groups of glucose tolerance. When comparing rates of MetS, these were lower in normoglycemic women of the intervention group compared to the standard-care group (7.9% vs. 12.9%, respectively, *p* < 0.05). Moreover, no significant differences were found when comparing rates of having from 0 to 6 components of MetS in neither GDM nor non-GDM women. However, there was a tendency showing women in the intervention group to have higher rates of having 0 components of MetS and lower rates of having ≥ 1, both when analyzing GDM and non-GDM women.

## 4. Discussion

This study reveals that an intervention starting early in pregnancy and maintained throughout this period is associated with a 26% lower risk of having MetS at three months postpartum. No differences were found with regards to insulin resistance rates. Moreover, when analyzed by subgroups of glucose tolerance, normoglycemic women in the intervention group had lower rates of MetS than the standard-care group. To the best of our knowledge, no other studies have compared the presence of MetS and IrS in early postpartum between women who received a nutritional intervention and controls.

It is difficult to make comparisons of these results with other studies because in most cases, the presence of MetS is evaluated after 12 months postpartum. Only Retnakaran et al. [17] have evaluated this in early postpartum. Their rates of MetS in normoglycemic women were similar to ours but lower in those with GDM. Their diagnosis of GDM was performed with the National Diabetes Data Group (NDDG) criteria in contrast to the International Association of Diabetes and Pregnancy Study Groups (IADPSG) criteria used in our study. Therefore, results between the two studies are difficult to compare. In addition, a prospective cohort study evaluated the presence of MetS postpartum in women with and without GDM, classified according to IADPSG criteria. They also found similar rates of MetS in GDM women and non-GDM women as ours [6]. Moreover, the prevalence of MetS in GDM women as compared to non-GDM women are much lower than those found by Noctor et al. [5]. However, the follow-up in their sample was of three years postpartum.

The lipid panel and blood pressure were better in the intervention group. Significant differences were observed mostly when comparing GDM women, where women in the intervention group had a more favourable profile. Dyslipidemia seems to be highly prevalent in both normoglycemic and GDM women in early postpartum (6–12 weeks) [18]. Screening for lipid abnormalities postpartum is not performed routinely, even in GDM women. This should be given consideration.

The intervention group had significantly lower levels of fasting glucose and HbA1c levels, even when subdividing them in groups by glucose tolerance. These findings were similar to another study that found less impaired glucose tolerance in women in the intervention group as compared to the standard-care group by oral glucose tolerance test, but found no differences in fasting glucose levels [11]. Unfortunately, data is limited since no other studies have evaluated the effect of an antenatal intervention on the prevention or delay of glucose abnormalities postpartum.

Overall, there was lower weight retention in the intervention group. Similar to this, Ferrara et al. [19] showed that an intervention starting in pregnancy and maintained in the postpartum period could prevent weight retention. Other authors have found no differences between groups [11]. However, their sample was women of high-risk (BMI ≥ 30 kg/m^2^). Thus, all women, both from the intervention and standard-care group, probably received counseling on weight loss strategies. Postpartum weight retention has been associated with a higher risk of recurrent GDM, type 2 diabetes, and cardiovascular diseases [20]. Therefore, returning to pregestational body weight is important, especially in women with a history of GDM.

The intervention group maintained better scores in pregnancy and postpartum than the standard-care group, as was found in the RADIEL study. According to their findings, women in the standard-care experienced a decline in their dietary habits while the intervention group maintained better dietary quality [11]. Postpartum dietary habits reveal that of women with GDM those in the intervention group had better dietary habits—as reflected by the scores—than the standard-care group. In the LIMIT trial, authors also found that antenatal lifestyle intervention was associated with persistence of improved postpartum dietary habits as compared to the standard-care group in overweight and obese women [21].

According to our results, all women benefit from an antenatal intervention, highlighting the importance of adopting healthy dietary habits in pregnancy. Women with and without GDM find barriers to adhere to a healthy lifestyle in the postpartum period, generally due to competing priorities such as childcare, tiredness, work duties, lack of time, and motivation [22,23]. It is mandatory to educate women on how important it is to attend the postpartum evaluation and maintain a healthy lifestyle, especially in those who have a history of GDM.

This study has some limitations. One weakness is that information and selection bias cannot be ruled out. The standard-care group included more women that had been diagnosed with GDM than the early-intervention group (49.5% versus 19.9%). This is because in the 2013–2015 period—when the standard-care group was included—a much higher proportion of women who had GDM attended a postpartum evaluation. Considering this, we tried to include women with similar characteristics in the postpartum evaluation. For instance, breastfeeding has been associated with improved insulin sensitivity postpartum in women with prior GDM [24]. Therefore, only women who were breastfeeding were included in this analysis in order to make women between groups as similar as possible. Also, results show that better metabolic outcomes in women in the intervention group persist even after subdividing by groups of glucose tolerance. A second limitation is that prepregnancy body weight is self-reported. However, this limitation is shared by most studies. Lastly, the groups in this cohort were not parallel groups, so participants from the two groups were included at different times. However, these results reflect the changes that have happened in the usual clinical practice throughout time. Therefore, these results could be reproduced in other centers where these women are treated.

Associations between the adherence to a MedDiet rich in extra virgin olive oil and nuts in pregnancy and improved materno-fetal events have been shown in women with and without GDM [7,8,9,12,13]. Results from these studies seem to indicate the benefits extend to the postpartum period.

## 5. Conclusions

In conclusion, this study reveals that early intervention in pregnancy seems to reduce the risk of developing MetS. It also seemed to decrease the rates of each MetS component, prediabetes, hypertension and lipid abnormalities, irrespective to having had GDM. Thus, all women should be evaluated in the fourth trimester, independent of having a history of GDM. Long-term implications should be studied further.

## Figures and Tables

**Figure 1 jcm-08-01499-f001:**
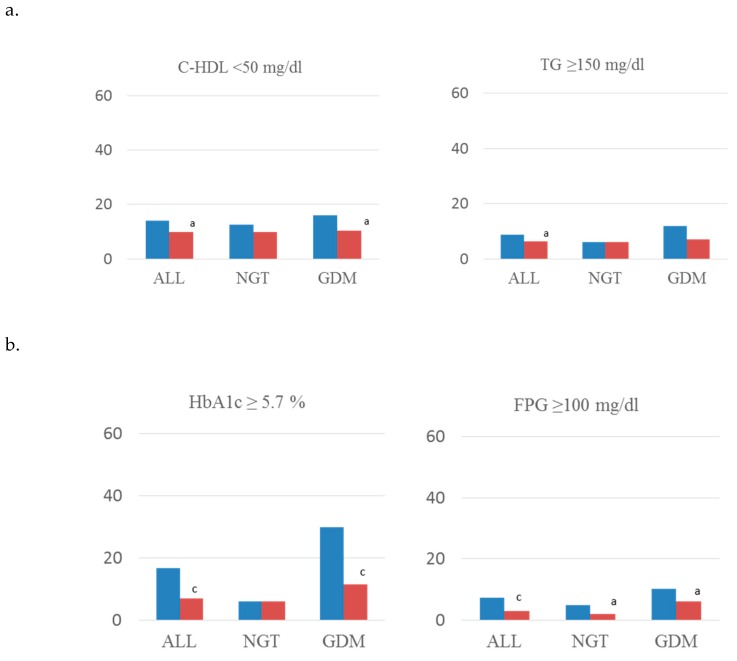
Rates of components of MetS concerning (**a**) lipid abnormalities, (**b**) glucose regulation, (**c**) blood pressure, and (**d**) anthropometric parameters. Standard-care group early intervention group. BMI, body mass index; WC, waist circumference; SBP, systolic blood pressure; DBP, diastolic blood pressure; C-HDL, HDL cholesterol; TG, triglycerides; FPG, fasting plasma glucose; GDM, gestational diabetes mellitus; NGT, normal glucose tolerance. a: *p* < 0.05; b: *p* < 0.01; c: *p* < 0.001.

**Figure 2 jcm-08-01499-f002:**
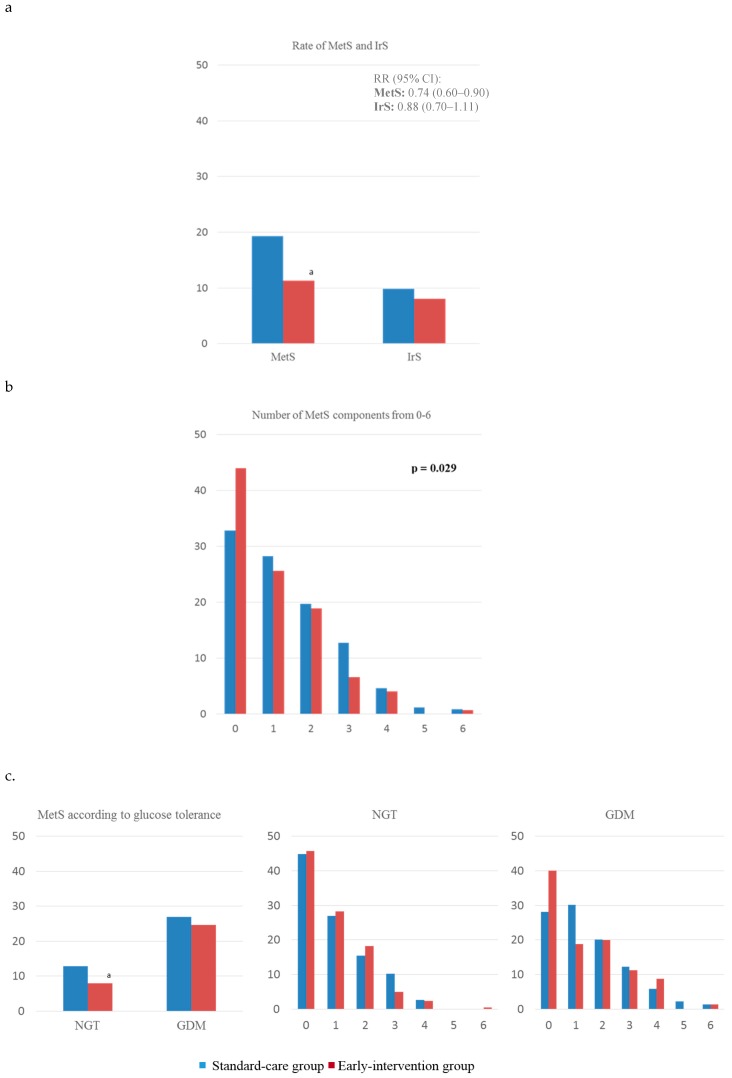
Presence of (**a**) complete MetS and IrS comparing standard-care and intervention groups and (**b**) of having from 0–6 components of MetS; and (**c**) complete MetS and of having 0–6 components of MetS by groups of glucose tolerance. MetS, metabolic syndrome; IrS, insulin resistance syndrome; GDM, gestational diabetes mellitus; NGT, normal glucose tolerance. a: *p* < 0.05; b: *p* < 0.01; c: *p* < 0.001.

**Table 1 jcm-08-01499-t001:** Demographic and clinical characteristics of the sample classified as standard care and early-intervention group.

Variables	Standard-Care Group (*n* = 676)	Early-Intervention Group (*n* = 999)	*p* Value
Age (years)	33.54 ± 5.11	33.32 ± 4.78	0.369
Race/Ethnicity n (%)			
Caucasian	467 (69.3)	676(67.7)	0.275
Hispanic	185 (27.4)	303 (30.4)	
Others	24 (1.9)	20 (1.9)	
Family history >1 component of MetS	216 (32.0)	455 (45.5)	0.001
Miscarriage	178 (26.4)	335 (33.5)	0.006
Number of pregnancies n (%)			
Primiparous	292 (43.8)	447 (44.8)	0.998
Second pregnancy	208 (31.2)	301 (30.1)	
>2 pregnancies	176 (26.0)	251 (25.1)	
Prepregnancy BW (kg)	63.3 ± 12.4	61.6 ± 11.1	0.003
Prepregnancy BMI (kg/m^2^)	24.2 ± 4.3	23.3 ± 4.0	0.001
BW at 12–14 GW (kg)	64.1 ± 11.7	63.5 ± 11.2	0.421
BW gain at 12–14 GW (kg)	2.0 ± 3.1	1.9 ± 3.1	0.338
BW gain at 24–28 GW	7.6 ± 3.8	7.0 ± 4.2	0.05
BW gain at 36–38 GW	11.4 ± 5.17	11.8 ± 6.4	
University Degree	442 (65.4)	709 (71.0)	0.209
Employed	543 (80.6)	632 (82.1)	0.089
Smoker			
Never	421 (62.6)	582 (58.3)	0.006
Current	142 (21.2)	166 (16.6)	
GDM at 24–28 GW	304 (49.5)	199 (19.9)	0.001

Data are mean ± SDM or number (%). MetS, metabolic syndrome; BW, body weight; BMI, body mass index; GW, gestational weeks; GDM, gestational diabetes mellitus.

**Table 2 jcm-08-01499-t002:** Nutrition and MedDiet scores before, during gestation, and at 12–14 weeks postpartum by groups of intervention and subdivided by glucose tolerance.

Variables	Standard-Care Group			Early-Intervention Group		
	All	Glucose Tolerance		All	Glucose Tolerance	
		NGT	GDM		NGT	GDM
Pregestational						
Nutrition Score	1.2 ± 3.4	0.4 ± 3.1	1.9 ± 3.5	0.5 ± 3.2 ***	0.5 ± 3.1	0.7 ± 2.8 ***
MedDiet Score	5.1 ± 1.8	5.0 ± 1.8	5.1 ± 1.7	5.4 ± 1.7 **	5.3 ± 1.7 *	5.5 ± 1.7
36–38 GW						
Nutrition Score	3.8 ± 3.6	2.6 ± 3.1	6.9 ± 2.9	4.7 ± 3.6 **	3.9 ± 3.3 ***	7.7 ± 3.0
MD (95% CI)	3.2 (2.7-3.8) ^c^	2.2 (1.6-2.8) ^c^	6.1 (5.1-7.1) ^c^	4.2 (3.9-4.5) ^c^	3.4 (3.2-3.7) ^c^	7.0 (6.4-7.6) ^c^
MedDiet Score	5.8 ± 1.8	5.1 ± 1.4	7.7 ± 1.4	6.4 ± 2.0 ***	5.9 ± 1.8 ***	8.1 ± 1.9
MD (95% CI)	0.8(0.4–1.1) ^c^	0.0 (−0.3–0.4)	2.8 (2.3–3.4) ^c^	1.0 (0.9–1.2) ^c^	0.7 (0.5–0.8) ^c^	2.6 (2.2–3.0)
Postpartum						
Nutrition Score	2.6 ± 3.6	2.4 ± 3.7	3.3 ± 3.5	3.1 ± 3.7	2.8 ± 3.7^*^	3.9 ± 3.5
MD (95% CI)	2.0 (1.5–2.6) ^c^	2.0 (1.4–2.6) ^c^	2.1 (1.1–3.1) ^c^	2.6 (2.3–2.8) ^c^	2.4 (2.1–2.7) ^c^	3.2 (2.6–3.8) ^c^
MedDiet Score	5.5 ± 1.6	5.5 ± 1.7	5.6 ± 1.5	6.0 ± 1.7 ***	5.8 ± 1.7 *	6.6 ± 1.7 **
MD (95% CI)	0.5 (0.1-0.8) ^b^	0.5 (0.1–0.8) ^a^	0.4 (−0.2–1.0)	0.6 (0.5–0.8) ^c^	0.5 (0.4–0.7) ^c^	1.0 (0.6–1.3) ^c^

Data are mean ± SDM or number (%). NGT, normal glucose tolerance; GDM, gestational diabetes mellitus; MedDiet, Mediterranean diet; GW, gestational week; MD, Mean differences with pregestational Score; CI, confidence interval. * *p* < 0.05; ** *p* < 0.01; and *** *p* < 0.001, denote differences between women in the standard-care and intervention groups. ^a^
*p* < 0.05; ^b^
*p* < 0.01; and ^c^
*p* < 0.001, denote differences with pregestational score.

**Table 3 jcm-08-01499-t003:** Postpartum clinical and laboratory data by groups of intervention and subdivided by glucose tolerance.

**Clinical and Biochemical Parameters**	**Variables**	**Standard-Care Group**			**Early-Intervention Group**		
		**All**	**Glucose Tolerance**		**All**	**Glucose Tolerance**	
			**NGT**	**GDM**		**NGT**	**GDM**
	N (%)	676	372 (55.0)	304 (45.0)	999	800 (80.1)	199 (19.9)
	Age (year)	33.5 ± 5.1	32.3 ± 5.3	34.3 ± 4.9	33.3 ± 4.8	33.0 ± 4.8	34.3 ± 4.5
Anthropometric parameters	Pregestational BW (kg)	63.3 ± 12.4	60.7 ± 10.2	65.2 ± 13.1	61.6 ± 11.1 **	60.6 ± 11.1	65.1 ± 12.7
	Pregestational BMI (kg/m^2^)	24.2 ± 4.3	23.0 ± 3.9	24.9 ± 4.4	23.3 ± 4.0 ***	22.9 ± 3.6	24.9 ± 4.6
	Postpartum BW (kg)	67.7 ± 12.8	65.5 ± 12.2	68.7 ± 13.2	66.3 ± 11.3	65. 5 ± 10.8	68.6 ± 11.9
	Postpartum BMI (kg/m^2^)	25.8 ± 4.5	24.9 ± 4.4	26.3 ± 4.6	25.1 ± 4.2 *	24.7 ± 4.0	26.3 ± 4.5
	BW change (kg)	4.8 ± 6.0	5.2 ± 4.8	3.0 ± 5.4	3.7 ± 5.4 **	5.3 ± 5.9	3.3 ± 5.2
	WC (cm)	85.7 ± 10.3	84.2 ± 9.2	89.5 ± 8.6	85.8 ± 9.1	84.6 ± 9.1	86.8 ± 10.2 *
Glucose regulation	Fasting glucose (mg/dL)	87.3 ± 9.1	84.2 ± 9.2	89.0 ± 8.6	84.4 ± 7.6 ***	83.7 ± 7.4	87.1 ± 7.8 *
	HbA1c-IFCC %	5.4 ± 0.3	5.3 ± 0.3	5.4 ± 0.3	5.3 ± 0.3 ***	5.2 ± 0.3	5.3 ± 0.3 ***
	Fasting insulin (mcUI/mL)	6.9 ± 6.0	7.4 ± 6.2	6.9 ± 5.6	6.5 ± 5.9	6.4 ± 5.9 *	6.9 ± 6.0
	HOMA-IR	1.7 ± 1.4	1.8 ± 1.6	1.7 ± 1.3	1.6 ± 1.4	1.6 ± 1.4	1.7 ± 1.4
Blood pressure	sBP (mm Hg)	116 ± 14	111 ± 13	121 ± 14	111 ± 12 ***	110 ± 12	114 ± 12 ***
	dBP (mm Hg)	73 ± 10	71 ± 11	76 ± 11	71 ± 10 *	70 ± 9	73 ± 9 *
Lipid panel	Total cholesterol (mg/dL)	211 ± 42	202 ± 38	214 ± 42	197 ± 37 ***	195 ± 36 *	203 ± 38 **
	HDL cholesterol (mg/dL)	63 ± 14	64 ± 13	62 ± 13	67 ± 18 ***	64 ± 19	68 ± 15 ***
	LDL cholesterol (mg/dL)	130 ± 36	124 ± 31	130 ± 37	116 ± 30 ***	114 ± 29 ***	123 ± 33
	Triglycerides (g/L)	88 ± 45	82 ± 41	89 ± 45	82 ± 47 **	80 ± 47	85 ± 47

Data are mean ± standard deviation or number (%). NGT, normal glucose tolerance; GDM, gestational diabetes mellitus; BW, body weight; BMI, Body mass index; WC, waist circumference; sBP, systolic blood pressure; dBP; diastolic blood pressure. * *p* < 0.05; ** *p* < 0.01; and *** *p* < 0.001, denote differences between both cohorts.

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
