# Peer review of "An Early, Universal Mediterranean Diet-Based Intervention in Pregnancy Reduces Cardiovascular Risk Factors in the “Fourth Trimester”"

_jcm, 2019, doi:10.3390/jcm8091499_

Round 1
Reviewer 1 Report
The aim of the prospective cohort study performed by Assaf-Balut was to evaluate whether an early nutritional intervention based on Mediterranean diet in pregnancy reduced 12-14 weeks postpartum the women’s cardiovascular risk, determined by the presence of metabolic syndrome and insulin resistance. The study compared women from the standard-care group, who received standard-care dietary guidelines, with the intervention group, who received Mediterranean diet (MedDiet)-based dietary guidelines, supplemented with extra-virgin olive oil and nuts. The study shows that an early MedDiet-based nutritional intervention in pregnancy is associated with 26% reduction in 4 moths postpartum rates of MetS.
This study is of interest, gives new information and has been well-performed. I have no negative comment.
Author Response
We are glad this paper has been of your interest! Thank you for your kind comment.
Reviewer 2 Report
An early, universal Mediterranean diet-based intervention in pregnancy reduces cardiovascular risk factors in the "fourth trimester".
Authors aimed to " to evaluate the presence of MetS and insulin resistance –as indicators63 of cardiovascular risk− at 12-14 weeks postpartum in women who followed a Mediterranean 64 diet-based nutritional intervention in early gestation (from 12 gestational weeks) compared to65 women who followed standard care. In addition, comparisons were made within groups of normoglycemic and women with GDM.
I found the paper very confusing and with several drawbacks.
Study Design and Methods:
The principal drawback is the study design. In Page 2, L80 authors state that "This is a prospective cohort study that includes women from three different studies[7,13,14]." In the following line they state "All women had to be breastfeeding at 12-14 weeks postpartum. If the study was a cohort study, then, it was a retrospective cohort.
But this is not the major drawback. In the statistical analysis section authors state: "The magnitude of association between study groups and binary outcomes (MetS and each components and IrS) were evaluated using the relative risk (RR) and 95% CI." This type of analysis is the right one for a cohort study; however I was not able to trace any results about Relative Risk.
Page 3, L86-112. The cohort composition in very confusing. There must be a brief and accurate description of the cohort composition.
In my opinion, authors should decide if this is a cohort study or an intervention study. If this is a cohort the appropriate statistical analysis should be applied by using one of the measures of Relative Risk. Furthermore, in the discussion should be stressed the weakness of the study as per selection and information bias.
Author Response
Thank you very much for your kind comments and constructive suggestions. We agree with every one of them. We have tried to make the corresponding changes to improve this article. Please note that when we quote lines of the paper, we are referring to the “tracked-changes” version.
The changes applied are as follows.
An early, universal Mediterranean diet-based intervention in pregnancy reduces cardiovascular risk factors in the "fourth trimester".
Authors aimed to " to evaluate the presence of MetS and insulin resistance –as indicators63 of cardiovascular risk− at 12-14 weeks postpartum in women who followed a Mediterranean diet-based nutritional intervention in early gestation (from 12 gestational weeks) compared to women who followed standard care. In addition, comparisons were made within groups of normoglycemic and women with GDM.
I found the paper very confusing and with several drawbacks.
Study Design and Methods:
The principal drawback is the study design. In Page 2, L80 authors state that "This is a prospective cohort study that includes women from three different studies[7,13,14]." In the following line they state "All women had to be breastfeeding at 12-14 weeks postpartum. If the study was a cohort study, then, it was a retrospective cohort.
In agreement with this, we have changed the word “prospective” to “retrospective” (line 86). These women were analyzed retrospectively, from the postgestational visit at 3 months postpartum and only those who were breastfeeding were included in this analysis.
We have added a phrase in “inclusion criteria” to clarify that only women who attended postpartum evaluation at 12-14 weeks postpartum and who were exclusively breastfeeding at this moment were included in this analysis (line 93-95).
But this is not the major drawback. In the statistical analysis section authors state: "The magnitude of association between study groups and binary outcomes (MetS and each components and IrS) were evaluated using the relative risk (RR) and 95% CI." This type of analysis is the right one for a cohort study; however I was not able to trace any results about Relative Risk.
There is a comment in the paper about the results of RR for MetS in line 30 (abstract) and lines 277. However, we agree that in the results section no information is shown in the form of tables or figures and we had not included the results for neither IrS nor each component of MetS. Therefore, we have added in figure 2.a. the results of RR for MetS and IrS (lines 300-302) and we have added this information in the text (line 278). We had not included it previously in the original paper to avoid repeating data.
Moreover, in terms of RR results for each component of MetS, we have not included the information because no significant results were found. We have added a phrase to clarify this point for readers (line 278-279).
Page 3, L86-112. The cohort composition in very confusing. There must be a brief and accurate description of the cohort composition.
In agreement with this, we have changed the description and tried to make it easier to understand. We have eliminated the previous description (lines 102-122) and substituted it for what we think is a better version. In doing so, we have subdivided the section of “study design” in two sections: study design (lines 68-101) and study population (lines 132-153).
In my opinion, authors should decide if this is a cohort study or an intervention study. If this is a cohort the appropriate statistical analysis should be applied by using one of the measures of Relative Risk. Furthermore, in the discussion should be stressed the weakness of the study as per selection and information bias.
We believe this is a cohort study.
With regards to the weaknesses of this study, since it’s not an interventional-randomized study we agree this makes it difficult to exclude some bias. For instance, the fact that more women in the standard-care group than the intervention group attended postpartum evaluation may imply some information and selection bias (as explained in Lines 364-367). However, we tried to compensate for this by trying to make both groups of women as similar as possible (lines 367-371). Notwithstanding, this might lead to both selection and information bias, and therefore we have added a phrase to this statement in lines 363-364.
In addition, the fact that the cohort did not include women simultaneously but rather at different periods is a limitation we had discussed in the original paper (lines 373-375). This is an important limitation that could also lead to some bias. However, we attempted to explain this limitation in lines 375-377.
Please, if you referred to other selection and information bias, let us know to make the required amendments.
Reviewer 3 Report
This was an interesting paper with a large data and comprehensive methods. Also the main result/conclusion that the early intervention in pregnancy seems to reduce the risk of developing MetS was important. This result supports the idea that dietary/lifestyle interventions early in pregnancy can have effective results for the future health of the women.
Author Response
Thank you for your kind comment.
Round 2
Reviewer 2 Report
The paper is now suitable for publication